# Revising BMI Cut-Off Points for Obesity in a Weight Management Setting in Lebanon

**DOI:** 10.3390/ijerph17113832

**Published:** 2020-05-28

**Authors:** Leila Itani, Dima Kreidieh, Dana El Masri, Hana Tannir, Leila Chehade, Marwan El Ghoch

**Affiliations:** Department of Nutrition and Dietetics, Faculty of Health Sciences, Beirut Arab University, P.O. Box 11-5020 Riad El Solh, Beirut 11072809, Lebanon; l.itani@bau.edu.lb (L.I.); d.kraydeyeh@bau.edu.lb (D.K.); dana.masri@bau.edu.lb (D.E.M.); hana.tannir@bau.edu.lb (H.T.); leilachadi95@hotmail.com (L.C.)

**Keywords:** adiposity, body composition, body fat, overweight

## Abstract

Obesity is defined by the World Health Organization (WHO) as a body mass index (BMI) ≥ 30 Kg/m^2^. This study aimed to test the validity of this BMI cut-off point for adiposity in a weight management clinical setting in Lebanon. This cross-sectional study of 442 adults of mixed gender, categorized by the WHO BMI classification, included: 66 individuals of normal weight, 110 who were overweight and 266 with obesity. The clinical sample was referred to the Outpatient Clinic in the Department of Nutrition and Dietetics at Beirut Arab University (BAU) in Lebanon. All participants underwent anthropometric evaluation. The gold standard for defining obesity was based on the National Institutes of Health (NIH)/WHO guidelines for total body fat percentage (BF%). The best sensitivity and specificity were attained to predict obesity, according to the receiver operating characteristic curve (ROC) analysis. The BMI cut-off point for predicting obesity in the clinical sample was nearly 31.5 Kg/m^2^, and more than 90% of individuals with obesity and cardiometabolic disease were above this cut-off point. In conclusion, this new BMI cut-off point, an obesity definition higher than suggested in Western populations, was demonstrated to have clinical usefulness. Obesity guidelines in Lebanon, therefore, need revising.

## 1. Introduction

Obesity is a growing health problem, with prevalence increasing continually worldwide [1]. It is considered a major risk factor for several medical indications (e.g., diabetes, cardiovascular diseases, and cancer) [2] and for psychosocial diseases (e.g., depression, eating disorders and health-related quality-of-life impairment) [3,4], which are all conditions that unavoidably increase rates of morbidity and mortality [5]. Accordingly, international guidelines advocated for a broad spectrum of weight loss interventions for clinical practice, including lifestyle modification programs [6], Pharmacologic treatment [7], and bariatric surgery [8].

The World Health Organization (WHO) defines obesity as an excessive fat deposition in the adipose tissue [9,10]; therefore, the classification of adiposity based on body fat (BF) quantification and assessment seems to be the most accurate [11]. To classify obesity in adults, however, the WHO mainly relies on a body mass index (BMI) cut-off point that has been derived mainly from Western populations (i.e., Europeans); a BMI ≥ 30 Kg/m^2^ indicates obesity [9]. Currently, this cut-off point is widely used in clinical and research settings to define obesity, since it is considered a simple and cheap tool for assessment [12], especially because it is also highly correlated with BF [12]. However, BMI classification has its limitations [13] for at least two main reasons. Firstly, despite the strong association between BMI and BF, the former is not able to discriminate the latter from lean body mass. In fact, in a group of individuals with the same BMI, different values of BF% may be recognized. Secondly, this cut-off point has been determined in certain populations (i.e., Western populations) and its validity is not a certainty in others, where available literature reports inconsistent evidence and raises the need to test the accuracy of evidence in several ethnic groups [14], foremost in those from North Africa and Middle East countries.

The current study, therefore, aims to investigate the extent to which the WHO BMI cut-off point for obesity classification, based on Western populations, is universal and valid in other populations, namely in a clinical setting of Lebanese patients with obesity, who are seeking treatment. Within this scope, the hypothesis is that the cut-off point in our population will vary from that proposed by the WHO.

## 2. Materials and Methods

In this cross-sectional diagnostic study that adhered to the statement, Strengthening The Reporting of OBservational Studies in Epidemiology (STROBE) (Appendix A), a total of 442 individuals were recruited including: (i) a clinical sample of 376 individuals with overweight or obesity (BMI ≥ 25.0 kg/m^2^) who sought weight loss treatment and referred by general practitioners to the Nutritional and Weight Management Outpatient Clinic in the Department of Nutrition and Dietetics (BAU) in Lebanon and (ii) a sample of 66 normal weight (18.5 ≥ BMI < 25 Kg/m^2^) individuals from the general community. Recruitment took place during the period between January 2015 and November 2019. Classification into normal, overweight, or obese was based on the National Institutes for Health (NIH)/WHO guidelines for BMI classification [9,10]. Participants were included in the clinical sample if they were aged ≥ 18 years, have a BMI ≥ 25.0 kg/m^2^ and had at least one weight loss responsive comorbidity (i.e., type 2 diabetes, cardiovascular disease, sleep apnoea, severe joint disease, or two or more risk factors) as defined by the Adult Treatment Panel III [15]. Excluded patients were either pregnant or lactating, taking medication that affects body weight, or presented with weight loss related to medical comorbidities, or severe psychiatric disorders. The 66 normal weight individuals were recruited from the general population through a randomized community-based email survey. The study was approved by the Institutional Review Board of the BAU (No. 2017H-0034-HS-R-0241), and all participants were included after giving written informed consent.

A questionnaire that retrieved information on social and demographic data (e.g., age, marital status, employment, level of education, etc.) was administered to all participants.

An electronic weighing scale (SECA 2730-ASTRA, Germany) and a stadiometer were used to measure body weight and height, respectively. BMI was calculated using the standard formula dividing body weight measured in kilograms by the square of the height in meters.

A segmental body composition analyzer (MC-780MA, Tanita Corp., Tokyo, Japan) [16] was used to measure body composition using a standardized technique. Participant age, gender, and height information were entered into the device. Then the participant was instructed to stand in a stable position barefooted. Separate reading for different body segments compositions was obtained. These readings are based on an algorithm incorporating impedance, age, and height to estimate the total and regional BF and fat-free mass (FFM) [16]. Age and gender-specific obesity cut-off points for BF% were suggested by Gallagher and colleagues [17]: Females: 20–39 years: BF% ≥ 39%40–59 years: BF% ≥ 40%60–79 years: BF% ≥ 42%Males: 20–39 years: BF% ≥ 25%40–59 years: BF% ≥ 28%60–79 years: BF% ≥ 30%

Cardiometabolic disease in this study is defined as the presence of any diseases, such as type 2 diabetes, cardiovascular diseases (coronary heart disease, stroke, transient ischaemic attack, and peripheral arterial disease) and dyslipidemia (lowered level of high-density lipoprotein cholesterol, and increased level of low-density lipoprotein cholesterol and triglycerides). These were self-reported to be experienced either simultaneously or separately.

### Statistical Analysis

Descriptive statistics are presented, including means and standard deviations or frequencies and proportions for continuous and categorical variables, respectively. The Student *t*-test was used to compare means, and the Chi squared test of independence was used for proportions. To test the diagnostic performance of BMI in detecting obesity status, Gallagher and colleagues’ definition for obesity based on age- and gender-specific BF% was used as a gold standard [17]. A classification analysis was done by calculating sensitivity and specificity and area under the curve (AUC) of the receiver operating characteristic curve (ROC). The criterion value of BMI with maximum sensitivity and specificity was selected for the BMI cut-off points. An AUC > 0.8 indicates an excellent discriminating ability [18]. The cut-off scores achieving 90% sensitivity, and their corresponding specificities were also calculated. 

All values were considered significant at *p* < 0.05. NCSS 12.0.2 (NCSS, NCSS, LLC. Kaysville, UT, USA) was used for the statistical analysis. A power analysis for the sample size was determined using PASS software (PASS 11. NCSS, LLC. Kaysville, UT, USA). For the sample of 202 patients classified with obesity versus 240 with an alpha of 0.05 and AUC of 0.965, the power is 1.000.

## 3. Results 

Table 1 shows the demographic characteristics of the study participants. The sample included 134 (30.3%) males and 308 (69.7%) females with a mean age of 34.7 ± 14.7 years and a mean BMI of 31.8 ± 6.2 Kg/m^2^. Almost two thirds (64.5%) of the participants were young (20–39 years) with a similar age distribution for males (68.7%) and females (62.7%). Participants were mostly unmarried (53.0%) and unemployed (59.8%), with females being more likely to be married (63.2% vs. 48.5%) and unemployed (70.3% vs. 35.8%) compared to males. Based on the WHO BMI classification [10], almost two thirds (60.2%) of the sample were classified with obesity (BMI ≥ 30 Kg/m^2^), with a similar distribution for males (57.5%) and females (61.4%). In contrast, only 45.7% of the sample were classified with obesity based on Gallagher and colleagues’ age- and gender-specific BF% classification [17], with a similar distribution for males (50.0%) and females (43.8%) (Table 1). 

Figure 1 shows a strong association between BMI and BF% in males (ρ = 0.895; *p* < 0.0001) and females (ρ = 0.669; *p* < 0.0001); however, for a determined BMI, a wide range of BF% values are evident in both genders. 

The results of the ROC analysis on the diagnostic performance are shown in Table 2 and Figure 2a,b. The AUCs for males (0.965, *p* < 0.001) and females (0.789, *p* < 0.001) indicate excellent and acceptable discriminating ability of BMI with a 97% and 79% chance respectively to detect [18,19] BF% defined obesity (Table 2). The optimal BMI cut-off points for discriminating males and females with obesity were 31.53 Kg/m^2^ and 31.44 Kg/m^2^, respectively. The BMI cut-off point for males achieved high sensitivity (85.1%) and specificity (85.1%), indicating a lower chance for false negatives and false positives. For females, the sensitivity (71.9%) and specificity (62.4%) were relatively lower. The cut-off point that achieved 90% sensitivity for males was 30.48 Kg/m^2^, with a relatively high specificity of 83.6%, which is close to the WHO cut-off point. However, for females, it was 28.85 Kg/m^2^ with a relatively low specificity of 46.8%, indicating a higher chance for false positives (Table 2). According to the determined cut-off points, obesity affected 50.0% of males and 52.6% of females, which is closer to Gallagher’s and colleagues’ definition [17] than to the WHO definition [10].

Finally, among individuals with a BMI ≥ 30 Kg/m^2^, 101 had cardiometabolic disease (34 males, 67 females). Of those, among males, 31(91.2%) had a BMI ≥ 31.53 Kg/m^2^, and only 3 (8.8%) had a BMI between 30 and 31.53 Kg/m^2^; among females, 60 (89.6%) had a BMI ≥ 31.44 Kg/m^2^, and only 7 (10.4%) had a BMI between 30 and 31.44 Kg/m^2^.

## 4. Discussion

The current study aimed to provide benchmark data on the validity of the WHO BMI cut-off point by determining adiposity status in a clinical setting in Lebanon for weight management treatment of adults of both genders with obesity.

The main finding was the identification of a new BMI cut-off point for obesity in a clinical sample of participants undergoing weight loss treatment. This new cut-off point (i.e., BMI nearly 31.5 Kg/m^2^) is significantly different from the WHO BMI cut-off point [10]. Several studies with a similar objective have been carried out on other populations [20,21,22,23,24]. Among these, some confirmed the suitability of the BMI cut-off point (i.e., ≥30 Kg/m^2^) suggested by the WHO to define obesity [22], while others did not [20,21,23,24]. Some studies found lower cut-off points [20,21,23] and another—in line with the current findings—found higher BMI cut-off points [24].

Moreover, in this study, almost 91% with both obesity and cardiometabolic disease scored above the newly established cut-off point (i.e., BMI ≥ 31.5 Kg/m^2^), which highlights the clinical implications of these findings: there is a need to adopt this cut-off point to define obesity in Lebanon. Moreover, a difference in BF% at different BMI values is reported for different populations [25]. As the population in the current is not European and might differ in BMI corresponding to different BF%, a BMI approaching 31.5 Kg/m^2^ is plausible to detect the need for clinical or public health action in the Lebanese population [25].

The current study has several strengths and limitations. The strengths reside in that it is one of the few studies to assess the validity of the WHO BMI cut-off point by determining the adiposity status in a relatively large group of treatment-seeking patients with obesity, in a ‘real-world’ clinical setting in an Arab-speaking country. However, several limitations are to be noted. First, the self-reported cardiometabolic disease information is not an objective clinical assessment. Second, the impedance analyzer is used for measuring body composition while it is still not accepted as a gold standard technique for patients with obesity; despite its validation versus dual-energy X-ray absorptiometry (DXA) scan [16]. Moreover, the clinical sample was based only on treatment-seeking outpatients at baseline of a weight loss treatment program. This bias prevents the findings from being representative of the population affected by obesity in all clinical settings. The findings in this study may not be extrapolated to patients with obesity seeking other treatment modalities (i.e., bariatric surgery, pharmacological interventions, etc.).

For this reason, three new directions for future research are needed. Firstly, research needs to confirm this cut-off point (i.e., 31.5 Kg/m^2^) in other groups of patients with obesity in Lebanon who are seeking other weight management treatments (i.e., bariatric surgery, pharmacological interventions, etc.). Secondly, on a national level, large sample studies are required to assess the validity of normal (≥18.5 Kg/m^2^) and overweight (≥25 Kg/m^2^) cut-off points and, if necessary, to determine new ones. Finally, similar studies to this one are needed in countries in the Middle East and North Africa region, which is known to have a high prevalence of obesity.

## 5. Conclusions

This study provides evidence that the optimal BMI cut-off point (i.e., ≥31.5 Kg/m^2^) corresponding to obesity in the population of the current study varies from that widely used (i.e., ≥30 Kg/m^2^). Therefore, it is recommended that this new cut-off point should be applied in clinical and public health settings for screening individuals for obesity in Lebanon. On a general scale, the ethnicity factor should be taken into account, where popular BMI cut-off point for classifying obesity determined in certain populations (i.e., Western populations) cannot be applied by default in others, and need to be tested initially for its validity and clinical usefulness.

## Figures and Tables

**Figure 1 ijerph-17-03832-f001:**
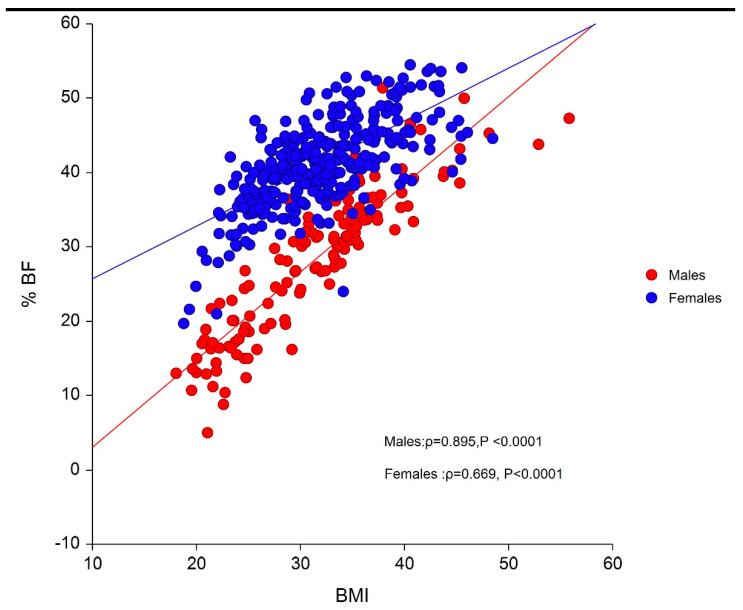
Association between BMI and BF% of the study sample (n = 442).

**Figure 2 ijerph-17-03832-f002:**
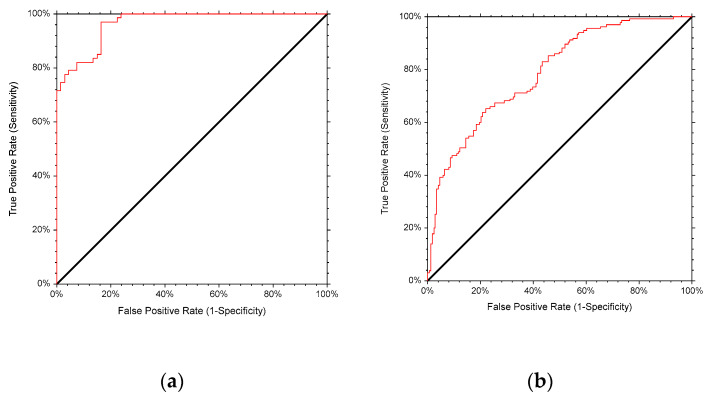
(**a**) ROC for the BMI cut-off point to detect obesity based on BF% as the gold standard in males; (**b**) ROC for the BMI cut-off point to detect obesity based on BF% as the gold standard in females.

**Table 1 ijerph-17-03832-t001:** Sociodemographic, anthropometric, and cardiometabolic characteristics of the study sample (n = 442) ^Ŧ^.

Demographics	Total N = 442	Males N = 134	Females N = 308	Significance
Age (years)	34.7 (14.7)	33.4(14.0)	35.3(15.0)	0.196
				X^2^ = 1.77; *p* = 0.412
20–39	285 (64.5)	92 (68.7)	193 (62.7)	
40–59	128 (29.0)	33 (24.6)	95 (30.8)	
60–80	29 (6.56)	9 (6.7)	20 (6.5)	
Marital status				X^2^ = 7.961; *p* = 0.0005
Unmarried	232 (53.0)	84 (63.2)	148 (48.5)	
Married	206 (47.0)	49 (36.8)	157 (51.5)	
Employment				X^2^ = 45.971; *p* < 0.0001
Unemployed	263 (59.8)	48 (35.8)	215 (70.3)	
Employed	177 (40.2)	86 (64.2)	91 (29.7)	
**Anthropometrics**				
Weight (Kg)	85.0 (18.7)	95.0 (23.2)	80.9 (14.1)	*p* < 0.0001
Height (cm)	163.7 (9.5)	174.3 (6.8)	159.1 (6.2)	*p* < 0.0001
BMI Kg/m^2^ ^£^	31.8 (6.2)	31.3 (7.3)	32.0 (5.6)	*p* = 0.298
				X^2^ = 26.676; *p* < 0.0001
Normal weight	66 (14.9)	36 (26.9)	30 (9.7)	
With overweight	110 (24.9)	21 (15.7)	89 (28.9)	
With Obesity	266 (60.2)	77 (57.5)	189 (61.4)	

BF (Kg)	32.3 (12.04)	28.4 (15.3)	34.0 (9.9)	*p* = 0.00014
BF% ^¥^	37.3 (9.5)	28.1 (9.7)	41.3 (6.0)	*p* < 0.0001
				X^2^ = 1.432; *p* = 0.231
Without obesity	240 (54.3)	67 (50.0)	173 (56.2)	
With obesity	202 (45.7)	67 (50.0)	135 (43.8)	
**Cardiometabolic disease**				X^2^ = 0.294; *p* = 0.588
No	319(73.2)	95 (71.4)	224 (73.9)	
Yes	117(26.8)	38 (28.6)	79 (26.1)	
**Diabetes**				X^2^ = 0.002; *p* = 0.967
No	376 (90.4)	114 (90.5)	262 (90.3)	
Yes	40 (9.6)	12 (9.5)	28 (9.7)	
**Dyslipidemia**				X^2^ = 0.032; *p* = 0.858
No	343 (82.3)	103 (81.8)	240 (82.5)	
Yes	74 (17.8)	23 (18.3)	51 (17.5)	
**Hypertension**				X^2^ = 0.001; *p* = 0.980
No	380 (87.2)	116 (87.2)	264 (87.1)	
Yes	56 (12.8)	17 (12.8)	39 (12.9)	

^Ŧ^ Values are Means (SD) or Medians (IQR) for continuous variables and n (%) for categorical variables. ^£^ WHO classification for BMI. ^¥^ Gallagher et al., 2000 [17].

**Table 2 ijerph-17-03832-t002:** Diagnostic performance of the BMI cut off points to detect obesity by sex groups.

Sex	N	AUC	95%CI	*P* Value	Sensitivity	Specificity	Cut Off	Specificity at 90% Sensitivity	Cut-off at 90% Sensitivity
Males	134	0.965	0.930–0.983	<0.0001	0.851	0.851	31.53	0.836	30.48
Females	308	0.789	0.733–0.833	<0.0001	0.719	0.624	31.44	0.468	28.85

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
