# Peer review of "Revising BMI Cut-Off Points for Obesity in a Weight Management Setting in Lebanon"

_ijerph, 2020, doi:10.3390/ijerph17113832_

Round 1

Reviewer 1 Report

Thank you for allowing me to review this manuscript. My comments/questions are below.to review this manuscript. My comments/questions are below.

General comments

The authors have tested the optimal BMI cut-off point for obesity in Lebanese population, comparing to the total body fat percentage. The authors described an optimal BMI cut-off point of 31.92 kg/m2. The authors argue that this new cut-off point should be applied in clinical and public health settings for screening of individuals for obesity in Lebanon.

  • Was the optimal BMI cut-off of 31.92 Kg/m2 obtained for all the sample? Did the authors tested if women (vs. men) have a different cut-off point?

  • All the references studies (20-23) described an optimal BMI for obesity in their population below 30kg/m2, and in this case it makes sense to use a cut-off point below to the WHO’s cut-off point, because we are failing to identify people in risk. However, I do not see the advantage in use a higher cut-off points to identify obesity, as the authors mentioned, 18% had also a cardiometabolic disease. Could the authors explain the advantages of using this new cut-off point?

I agree that BMI have several problems in identify with accuracy people with obesity, however I do not believe that using a higher cut-off point would be the solution. We need to use another criteria beyond BMI; in 2019 the EASO issue ‘The ABCD of Obesity’ (Obes Facts. 2019;12(2):131-136.), describing obesity as an ‘adiposity-based chronic disease’ (ABCD), proposing new criteria for diagnostic criteria including ethology, degree of adiposity, and health risk.

Author Response

Reviewer 1

General comments

The authors have tested the optimal BMI cut-off point for obesity in Lebanese population, comparing to the total body fat percentage. The authors described an optimal BMI cut-off point of 31.92 kg/m2. The authors argue that this new cut-off point should be applied in clinical and public health settings for screening of individuals for obesity in Lebanon.

Was the optimal BMI cut-off of 31.92 Kg/m2 obtained for all the sample? Did the authors tested if women (vs. men) have a different cut-off point?

Response: Initially the optimal cut-off point was obtained for the entire sample. However according to the reviewer's comment the analysis was re-conducted stratified by sex and reported in the abstract and manuscript as well as tables 1 and 2 and figure 1 and 2.

All the references studies (20-23) described an optimal BMI for obesity in their population below 30kg/m2, and in this case it makes sense to use a cut-off point below to the WHO’s cut-off point, because we are failing to identify people in risk. However, I do not see the advantage in use a higher cut-off points to identify obesity, as the authors mentioned, 18% had also a cardio-metabolic disease. Could the authors explain the advantages of using this new cut-off point?

Response: Our study is not the one study that found an optimal BMI for obesity in their population above 30kg/m2 (Almajwal AM, et al. Ann Saudi Med. 2009;29:437–445). The advantage of higher cut-off points to identify obesity stems in figuring out the false positives in terms of adiposity (i.e. BF%) and cardio metabolic diseases (<10%). We added this in the discussion section as well as conclusion with appropriate references (Lines 161-172).

I agree that BMI have several problems in identify with accuracy people with obesity, however I do not believe that using a higher cut-off point would be the solution. We need to use another criteria beyond BMI; in 2019 the EASO issue ‘The ABCD of Obesity’ (Obes Facts. 2019;12(2):131-136.), describing obesity as an ‘adiposity-based chronic disease’ (ABCD), proposing new criteria for diagnostic criteria including ethology, degree of adiposity, and health risk.

Response: We agree with the reviewer, now we mentioned the raised issue in the Conclusion section (Lines) and adding the appropriate reference (Lines 196-199).

Reviewer 2 Report

Abstract: The abstract should be a total of approximately 200 words maximum.  Please, reduce the number of words.

Keywords: The keywords must not be the same as those in the title and should be ordered alphabetically.

Introduction:

  • Indicate hypothesis raised
  • Indicate whether any studies with similar objectives have been carried out or similar questions have been asked in other places.
  • The importance of conducting this study should be reflected

Methods:

  • The methodology should indicate the type of study design being carried out.
  • The study must follow its corresponding scale of methodological quality. In this case, as it is a study of diagnostic tests, it must follow “Strengthening The Reporting of OBservational Studies in Epidemiology”: STROBE statement.
  • Line 59: Misplaced appointments 9 and 10.

Results:

  • The explanation of the tables and figures (especially table 2 and figure 2) are very brief. It should be more complete for better interpretation.

Discussion:

  • The results obtained should be discussed in the widest possible context, as this is done very briefly.
  • In addition to the limitations, indicate what biases the study has been exposed to.
  • Future research directions can be mentioned.

Conclusion:

  • Line 162: No information should be given in the first person.
  • The conclusion should be redrafted: It should be clear and concise, explaining in a generalized way the results obtained.
  • Future research is expressed in a broad way so it would be convenient to include it in the discussion. You can indicate something in the conclusion but in a brief way

References:

  • In the text, reference numbers should be placed in square brackets [].They must be changed as they appear in parentheses
  • The abbreviated name of the journals and the volume should be written in italics.
  • The year of the article must appear in bold

General commentary for authors

The study is very well approached and written, only some aspects must be added and corrected to obtain a greater quality of the same one. In addition, attention should be paid to the indications of the references.

Author Response

Reviewer 2

Abstract: The abstract should be a total of approximately 200 words maximum. Please, reduce the number of words.

Response: Done as suggested now exactly 200 (Lines 10-23).

Keywords: The keywords must not be the same as those in the title and should be ordered alphabetically.

Response: Done as suggested, now are different from those in title and ordered in alphabetical order (Line 24).

Introduction:

Indicate hypothesis raised

Response: Done as suggested, now we added a hypothesis (Lines 50-52).

Indicate whether any studies with similar objectives have been carried out or similar questions have been asked in other places.

Response: Done as suggested (Lines 45-47 and 161-165).

The importance of conducting this study should be reflected

Response: Done as suggested (Line 41-47).

Methods:

The methodology should indicate the type of study design being carried out.

Response: Done as suggested (Line 54).

The study must follow its corresponding scale of methodological quality. In this case, as it is a study of diagnostic tests, it must follow “Strengthening The Reporting of OBservational Studies in Epidemiology”: STROBE statement.

Response: Done as suggested (Line 55).

Line 59: Misplaced appointments 9 and 10.

Response: Done as suggested (Lines 62).

Results:

The explanation of the tables and figures (especially table 2 and figure 2) are very brief. It should be more complete for better interpretation.

Response: Done as suggested (Lines 114-121 and 132-147).

Discussion:

The results obtained should be discussed in the widest possible context, as this is done very briefly.

Response: Done as suggested (Lines 161-173).

In addition to the limitations, indicate what biases the study has been exposed to.

Response: Done as suggested (Lines 180-184).

Future research directions can be mentioned.

Response: Done as suggested (185-191).

Conclusion:

Line 162: No information should be given in the first person.

Response: Changes according to the reviewer’s suggestion (Line 193).

The conclusion should be redrafted: It should be clear and concise, explaining in a generalized way the results obtained.

Response: Done as suggested (Lines 196-199).

Future research is expressed in a broad way so it would be convenient to include it in the discussion. You can indicate something in the conclusion but in a brief way

Response: Done as suggested (Lines 185-191).

References:

In the text, reference numbers should be placed in square brackets []. They must be changed as they appear in parentheses. The abbreviated name of the journals and the volume should be written in italics. The year of the article must appear in bold

Response: Done as suggested in the text and Reference section.

General commentary for authors

The study is very well approached and written only some aspects must be added and corrected to obtain a greater quality of the same one. In addition, attention should be paid to the indications of the references.

Response: Done as suggested.

Round 2

Reviewer 2 Report

Abstract:

  • The first person appears. A scientific article should not be written in the first person, it should always be written in an impersonal manner. (Line 11)

Introduction:

  • The first person appears. A scientific article should not be written in the first person, it should always be written in an impersonal manner. (Line 51)

Results:

  • The explanation of lines 133 to 136 could be improved for easier understanding

Discussion:

  • If several studies are mentioned, more than one should be referenced. (Line 162 "others in line with our findings - found higher BMI cut-off points [24].)

  • The first person appears. A scientific article should not be written in the first person, it should always be written in an impersonal manner. (Line 156, 162 , 163, 165, 167 and170)

Conclusion:

  • The first person appears. A scientific article should not be written in the first person, it should always be written in an impersonal manner. (Line 191 and 193)

  • The conclusion should not be quoted, this section should be a unique conclusion of the results of the manuscript.

  • The conclusion should be redrafted: It should be clear and concise, explaining in a generalized way the results obtained.

Author Response

Abstract:

The first person appears. A scientific article should not be written in the first person, it should always be written in an impersonal manner. (Line 11)

Response: The first person statement has been modified as suggested. (Line 11)

Introduction:

The first person appears. A scientific article should not be written in the first person, it should always be written in an impersonal manner. (Line 51)

Response: The first person statement has been modified as suggested. (Line 51)

Results:

The explanation of lines 133 to 136 could be improved for easier understanding

Response: Now the explanation in the indicated lines has been improved for better understanding. (Lines 134-135)

Discussion:

If several studies are mentioned, more than one should be referenced. (Line 162 "others in line with our findings - found higher BMI cut-off points [24].)

Response: We corrected the statement according to the reviewer comment as: and another – in line with the current findings - found higher BMI cut-off points [24]. (Line 162)

The first person appears. A scientific article should not be written in the first person, it should always be written in an impersonal manner. (Line 156, 162, 163, 165, 167 and170)

Response: The first person statements have been modified as suggested. (Line 156, 162, 163, 165, 167 and170)

Conclusion:

The first person appears. A scientific article should not be written in the first person, it should always be written in an impersonal manner. (Line 191 and 193)

Response: The first person statements have been modified as suggested. (Lines 191 and 192)

The conclusion should not be quoted this section should be a unique conclusion of the results of the manuscript.

Response: We remove the quoted part as well as its citation.

The conclusion should be redrafted: It should be clear and concise, explaining in a generalized way the results obtained.

Response: The conclusion now has been redrafted: to appear clear and concise, explaining in a generalized way the results obtained. (Lines 193-196)